# Effects of Dietary β-Glucan Feeding Strategy on the Growth, Physiological Response, and Gut Microbiota of Pacific White Shrimp, *Litopenaeus vannamei*, under Low Salinity

**DOI:** 10.3390/ani13243778

**Published:** 2023-12-07

**Authors:** Yanbing Qiao, Fenglu Han, Kunyu Lu, Li Zhou, Artur Rombenso, Erchao Li

**Affiliations:** 1School of Life Sciences, East China Normal University, Shanghai 200241, China; yanbing_q@163.com; 2Key Laboratory of Tropical Hydrobiology and Biotechnology of Hainan Province, Hainan Aquaculture Breeding Engineering Research Center, College of Marine Biology and Aquaculture, Hainan University, Haikou 570228, China; l15265951224@163.com (K.L.); li.zhou@sjtu.edu.cn (L.Z.); 3CSIRO, Agriculture and Food, Livestock & Aquaculture Program, Bribie Island Research Centre, Bribie Island, QLD 4507, Australia; artur.rombenso@csiro.au

**Keywords:** *Litopenaeus vannamei*, β-glucan, feeding strategies, low salinity, gut microbiota

## Abstract

**Simple Summary:**

With the rapid development of shrimp aquaculture, freshwater aquaculture production in China accounted for 35% of the total aquaculture production for *Litopenaeus vannamei*. However, the negative effects of low salinity stress also limit the production of shrimp. Therefore, the use of prebiotics is an effective strategy to alleviate these adverse effects. This study investigated the effects of different dietary β-glucan application frequencies on the growth performance, physiological response, and gut microbiota of Pacific white shrimp under low salinity conditions. The results indicated that the feeding pattern of dietary 0.1% β-glucan for 2 days and basic feed for 5 days per week could improve the antioxidant capacity, immune response, and intestinal health of *L. vannamei*.

**Abstract:**

An eight-week feeding trial was conducted to investigate the effects of a dietary β-glucan application strategy on the growth performance, physiological response, and gut microbiota of Pacific white shrimp (*Litopenaeus vannamei*) (0.49 ± 0.17 g) under low salinity. Six feeding strategies were established, including a continuous β-glucan-free diet group (control), a continuously fed group with a 0.1% β-glucan diet (T1), and groups with the following intermittent feeding patterns: 1 day of β-glucan diet and 6 days of β-glucan-free diet (T2), 2 days of β-glucan diet and 5 days of β-glucan-free diet (T3), 3 days of β-glucan diet and 4 days of β-glucan-free diet (T4), and 4 days of β-glucan diet and 3 days of β-glucan-free diet (T5) each week. No significant differences in growth performance among all the groups were found, although the condition factor was significantly higher in the T3 group than in the T1 and T5 groups (*p* < 0.05). The T-AOC and GPX activities were significantly lower in the T3 group than in the control group (*p* < 0.05). The MDA content was also significantly lower in the T2 group than in the T3 and T4 groups (*p* < 0.05). Additionally, the mRNA expression of the *Pen3a* gene was significantly upregulated in the hepatopancreas of the T4 group compared to the control and T5 groups (*p* < 0.05), and the *Toll* gene was also significantly upregulated in the T3 group compared to the T1 and T2 groups (*p* < 0.05). Dietary β-glucan induced changes in the alpha diversity and composition of the gut microbiota in different feeding strategies. The beta diversity of the gut microbiota in the T2 group was significantly different from that in the control group. The results of a KEGG analysis showed that gut function in the carbohydrate metabolism, immune system, and environmental adaptation pathways was significantly enhanced in the T3 group. These findings provide evidence that the intermittent feeding strategy of β-glucan could alleviate immune fatigue, impact antioxidant ability, and change gut microbiota composition of *L. vannamei* under low salinity.

## 1. Introduction

The Pacific white shrimp (*Litopenaeus vannamei*) is a globally cultivated and economically significant crustacean. This species has gained widespread popularity due to its strong viability, rapid growth, delicious taste, and rich nutrient content [1,2]. In 2018, *L. vannamei* represented 82.8% of the total crustacean production, amounting to 4.9 million tons [3]. There has been increasing interest in cultivating *L. vannamei* in low-salinity environments, driven by its potential for inland aquaculture development [4]. Nevertheless, environmental stresses are regarded as a hindrance to the advancement of inland aquaculture [5]. Shrimp raised in low salinity conditions have been experiencing adverse effects, including decreased specific growth, compromised immune function, and reduced stress resistance [6,7]. Hence, enhancing the ability of shrimp to withstand environmental stress is the primary approach for fostering growth and yield in low salinity conditions.

Nutritional enhancement has been demonstrated to activate the innate system, thereby enhancing the resistance of aquatic animals and improving their overall health [8,9,10,11,12]. There has been a greater focus on immunopreventive methods aimed at enhancing the resistance of organisms in aquaculture by bolstering the vitality of their immune system [13]. Several studies have reported that various immunostimulants can increase the acquired resistance of aquatic animals, including *L. vannamei* and grouper (*Epinephelus fuscoguttatus*) [14,15,16]. Natural immune-stimulating polysaccharides, like β-glucan, inulin, fructooligosaccharides, and mannan oligosaccharides, have found extensive application in shrimp aquaculture [17,18,19,20]. β-(1,3)-glucan, a natural polymer isolated from the cell wall of yeast and mold, is among the most effective immunostimulants [21,22,23]. It has been successfully employed in Nile tilapia (*Oreochromis niloticus*), *L. vannamei*, red sea bream (*Pagrus major*), and zebrafish (*Danio rerio*) [8,17,24,25,26]. These investigations have shown that dietary β-glucan could enhance growth, antioxidation, immune response, and metabolic regulation in aquatic species [27,28,29].

Currently, immunostimulants are extensively used and administered over an extended period during aquaculture. Dietary strategies for oral supplementation play a critical role in enhancing efficiency, cutting costs, and minimizing feed waste [30]. Improper administration of immunostimulants may render no growth benefits and even lead to immune fatigue that eventually provides less protection from infection by potent pathogens [31,32,33,34,35]. Intermittent administration has shown positive effects [36,37]. For example, dietary supplementation with *Ampithoe* sp. and fructooligosaccharide at various levels and feeding strategies has positively impacted the growth and antioxidant capacity of *L. vannamei* and blunt snout bream (*Megalobrama amblycephala*) [30,38]. The effectiveness of dietary β-glucan is related to the duration of its use, with feed strategies sometimes being short-term [39]. The continuous application of β-glucan for 40 days elevated the respiratory burst on the 24th day in black tiger shrimp (*Penaeus monodon*). Nevertheless, the respiratory burst decreased compared to the diet without β-glucan [40]. Similarly, long-term use of β-glucan could result in a nonreactive physiological status and poor immune response in rainbow trout (*Oncorhynchus mykiss*) [41]. Therefore, intermittent feeding with β-glucan supplementation could be a practical approach to prevent immune fatigue and sustain effectiveness. However, research on the effect of dietary β-glucan application strategies on the growth and immunity of aquatic animals under low salinity is limited.

Thus, this study aims to evaluate the impact of different feeding strategies, including continuous feeding and intermittent feeding with β-glucan supplementation, on the physiology, immune response, and gut microbiota of *L. vannamei* under low salinity.

## 2. Materials and Methods

### 2.1. Experimental Diets

Two isonitrogen and isolipidic practical diets were formulated with and without doses of 0.1% β-glucan. The experimental diet formula was prepared according to previous research [25]. Basically, the ingredients were thoroughly crushed and mixed, and then distilled water and oil were added to obtain a stiff dough. The diet was passed through pellets in a laboratory pellet mill using a 2 mm die (F-26, SCUT, industrial factory, Guangdong, China). After drying at 37 °C and being kept away from light, the diets were crushed, sieved with appropriate sieves, and stored in plastic containers at −20 °C until use. The diet formulation was smashed and analyzed for crude protein, crude lipid, and crude ash as mainly described by the methods of AOAC [42]. The dietary moisture was determined by drying the samples at 105 °C to a constant weight. The crude protein was determined using the Dumas combustion method [43]. The crude lipid levels were measured after diethyl ether extraction using the Soxhlet method (Buchi 36,680, Flawil, Switzerland). The ash content was determined after combustion in a muffle furnace at 550 °C for 10 h. The proximate composition is presented in Table 1.

### 2.2. Experimental Design, Growth Trial, and Sampling

*L. vannamei* postlarvae (PL 5) were obtained from Blue Ocean Biotechnology Co., Ltd. (Wenchang, Hainan, China). After being transported to the experimental facilities, the shrimp were relocated to a 250 L round tank and raised over three weeks. During this period, the salinity was gradually decreased from salinity 30 to salinity 3 by continuously adding fully aerated water to obtain low-salinity water. The postlarvae were fed commercial feed (protein 48%, lipid 5%, Alpha Feed Co., Ltd., Shenzhen, China) at 4% of their larval weight every day to meet their nutritional needs. One-third of the seawater was replaced daily. After that, the shrimp (0.49 ± 0.17 g) were randomly selected and stocked into twenty-four fiberglass circular tanks (diameter 80 cm and depth 150 cm) with a stocking density of 20 shrimp per tank. The experiment was divided into six treatment groups, including the control (the shrimp were fed a β-glucan-free diet), T1 (the shrimp were fed a β-glucan diet every day), T2 (the shrimp were fed the β-glucan diet for 1 day and the β-glucan-free diet for the next 6 days every week), T3 (the shrimp were fed the β-glucan diet for 2 days and the β-glucan-free diet for the next 5 days every week), T4 (the shrimp were fed the β-glucan diet for 3 days and the β-glucan-free diet for the next 4 days every week), and T5 (the shrimp were fed the β-glucan diet for 4 days and the β-glucan-free diet for the next 3 days every week) groups. This strategy continued throughout the eight-week feeding trial. There were four replicates for each group. During the growth trial, the shrimp were fed four times (07:00, 12:00, 17:00, and 24:00) daily, and the daily feed intake was 6–10% of the shrimp’s body weight to ensure apparent satiation. During the trial, the water quality parameters were carefully monitored to ensure optimal conditions for shrimp growth and survival. The water temperature (27 ± 0.5 °C), dissolved oxygen (>6.0 mg/L), pH (7.8 ± 0.5), and NH3-N (<0.1 mg/L) were monitored twice a week, and the photoperiod was 12 h/12 h (light: dark).

At the end of the trial, the shrimp were starved for 24 h. Then, all the shrimp were anesthetized using shattered ice; the tanks were counted and individually weighed to determine the survival, the specific growth rate, and the weight gain, and the length of each shrimp was recorded for the condition factor. Fourteen shrimp from each tank were randomly sampled in a sterile environment, and the hepatopancreas was separated and weighed for the antioxidant enzyme analysis, gene expression, and hepatosomatic index. In addition, the mid-gut of the shrimp was separated for the gut microbiota analysis. All the samples were directly placed in freezer tubes and stored at −80 °C before the subsequent analysis. Each sample was analyzed in duplicate. The parameters were calculated using the following formulae:Weight gain (%) = (final weight (g) − initial weight (g))/initial weight (g) × 100
Specific growth rate (% day^−1^) = (Ln final weight − Ln initial weight)/day × 100
Condition factor (%) = final weight(g)/(final body length, cm)^3^ × 100
Hepatosomatic index (%) = hepatosomatic weight/final weight × 100
Survival (%) = final number of shrimp/initial number of shrimp × 100

### 2.3. Antioxidant Enzyme Activities

Two shrimp hepatopancreas samples per tank were randomly obtained for weighing, and saline was added at a ratio of 1:9 before homogenization. The homogenate was centrifuged (10 min, 2500× *g*, and 4 °C), and the resulting supernatant was used for the enzyme activity analysis. These samples of hepatopancreas were used to measure the activities of total antioxidative capacity (T-AOC, determined using the FRAP method, code: A015-3-1), glutathione peroxidase (GPX, measured using a colorimetric method, code: A005-1-2), and the content of malondialdehyde (MDA, determined using the TBA method, code: A003-1-2) using commercial kits (Nanjing Jiancheng Institute, Nanjing, China). The procedures used for these biochemical enzyme assays were performed as previously described [44,45].

The total antioxidant capacity (T-AOC) in the hepatopancreas was quantified using the FRAP method with the T-AOC assay kit [46]. A 10% homogenate supernatant of the hepatopancreas was combined with the working liquid. Following a 6 min incubation period, the absorbance of the resulting mixture was measured at 405 nm. The glutathione peroxidase (GPX) activity in the hepatopancreas was assessed using the colorimetric method with the GPX assay kit. A 5% homogenate supernatant of the hepatopancreas was prepared to maintain an inhibition rate within the range of 45% to 55%. The supernatant was then mixed with the provided reagents according to the instructions provided with the kit. Subsequently, the absorbance of the resulting mixture was measured at 412 nm. The malondialdehyde (MDA) content in the hepatopancreas was quantified using the thiobarbituric acid (TBA) method with the MDA assay kit [47]. A 10% homogenate supernatant of the hepatopancreas was prepared and mixed with reagent 1 as well as working liquids 2 and 3. The resulting mixture was subjected to heat treatment at 95 °C in a water bath for 40 min. Subsequently, the tubes were cooled using running water and centrifuged at 4000 rpm at 4 °C for 10 min. A 400 μL portion of the supernatant was transferred to a microplate, and the absorbance was measured at 532 nm.

### 2.4. Gene Expression Analysis

Two shrimp hepatopancreas samples per tank were individually obtained to analyze gene expression. The total RNA of the hepatopancreas was extracted using the TRIzol Reagent (Invitrogen, Carlsbad, CA, USA) according to the instructions. Subsequently, genomic DNA contamination was removed by treatment with recombinant DNase. The quality and concentration of the RNA were determined using a NanoDrop 2000 system (Thermo Fisher Scientific, Waltham, MA, USA). cDNA was synthesized from the RNA using a RevertAid First Strand cDNA Synthesis Kit (Thermo Fisher Scientific, Waltham, MA, USA). After that, the cDNA was stored at −20 °C until further analysis.

A real-time quantitative PCR was carried out with a Biosystems 7500 Real-Time qPCR System (ABI, Waltham, MA, USA) in duplicate. The primer sequences for the genes of the hepatopancreas are detailed in Table 2. The PCR experimental procedure was performed according to a previous procedure [48]. Specifically, the PCRs were conducted using a reaction volume of 10 μL, which consisted of 0.5 μL of 10 mM forward and reverse primers each (1 mmol/L), 2.5 μL of diluted cDNA (1:5 dilution), and 5 μL of 2 × SYBR Premix Ex TaqTM. The reaction was programmed with the following steps: an initial denaturation at 94 °C for 3 min, followed by 40 cycles of denaturation at 94 °C for 15 s, annealing at 58 °C for 50 s, and extension at 72 °C for 20 s. The efficiency of each primer was tested using standardization curves with various dilutions of cDNA, subsequently validating the efficiency of the primers’ performance. The reference gene was normalized to *EF1α* and calculated using the 2^−ΔΔCt^ method [49].

### 2.5. Gut Microbiota Analysis

Three shrimp gut samples were individually obtained, one sample per tank, for analyzing the gut microbiota. Total genomic DNA samples were extracted using the OMEGA Soil DNA Kit (M5635-02) (Omega Bio-Tek, Norcross, GA, USA), following the manufacturer’s instructions. The quantity and quality of the extracted DNAs were measured using a NanoDrop NC2000 spectrophotometer (Thermo Fisher Scientific, Waltham, MA, USA) and agarose gel electrophoresis, respectively. PCR amplification of the bacterial 16S rRNA genes V3–V4 region was performed using the forward primer 338F (5′-ACTCCTACGGGAGGCAGCA-3′) and the reverse primer 806R (5′-CGGACTACHVGGGTWTCTAAT-3′). The PCR amplicons were purified with Vazyme VAHTSTM DNA Clean Beads (Vazyme, Nanjing, China,) and quantified using the Quant-iT PicoGreen dsDNA Assay Kit (Invitrogen, Carlsbad, CA, USA). After the individual quantification step, the amplicons were pooled in equal amounts, and pair-end 2250 bp sequencing was performed using the Illlumina NovaSeq platform with the NovaSeq 6000 SP Reagent Kit (500 cycles) at Shanghai Personal Biotechnology Co., Ltd. (Shanghai, China). Microbiome bioinformatics were performed with QIIME2 2019.4 and R packages (v3.2.0). Briefly, the raw sequence data were demultiplexed using the demux plugin followed by primers’ cutting with the cutadapt plugin. The sequences were then quality-filtered, denoised, merged, and chimera-removed using the DADA2 plugin. The non-singleton amplicon sequence variants were aligned with mafft and used to construct a phylogeny with fasttree2. The alpha diversity index values were determined using the ASV table in QIIME2, and statistical analyses were expressed as the mean SE and analyzed using two-way ANOVA (SPSS 23.0) with the feeding strategies, followed by multiple comparisons using Duncan’s test. A beta diversity analysis was performed to investigate the structural variation of the microbial communities across the samples using Jaccard metrics. The taxonomy compositions and abundances were visualized using MEGAN (https://www.wsi.uni-tuebingen.de/lehrstuehle/algorithms-in-bioinformatics/software/megan6/, accessed on 5 December 2023) and GraPhlAn (https://huttenhower.sph.harvard.edu/GraPhlAn, accessed on 5 December 2023). All the raw sequences were deposited in the NCBI Sequence Read Archive under accession number PRJNA846576.

### 2.6. Statistical Analysis

Prior to conducting statistical analyses, we rigorously assessed all the raw data for normality and homoscedasticity. These assessments were carried out using standard statistical methods appropriate for each data type. Following this, all the experimental results were statistically analyzed using IBM SPSS 22.0 (Chicago, IL, USA). The data were analyzed using a one-way analysis of variance (one-way ANOVA). Duncan’s multirange test was used to test the significance of the differences between the experiments. All the data are presented as the mean ± standard error (S.E.), and the statistically significant differences were set as *p* < 0.05.

## 3. Results

### 3.1. Growth Performance

The shrimp growth performance of the different groups after eight weeks is shown in Table 3. The shrimp condition factor of the T3 group was significantly higher than that of the shrimp from the T1 group (*p* < 0.05). No differences were found in all the other growth parameter among all the groups (*p* > 0.05).

### 3.2. Antioxidant Capacity

Compared to the control group, significantly lower activities of T-AOC in the shrimp hepatopancreas were found in all the groups that had been fed β-glucan diets, regardless of the feeding strategies (*p* < 0.05), except for group T2 (*p* > 0.05), while a significant lower GPX activity was only found in the shrimp of group T2 (Figure 1). No significant differences (*p* > 0.05) were observed among the MDA contents between the control and other groups; however, the shrimp in the T2 group had a significantly lower MDA content than the shrimp in both the T3 and T4 groups (*p* < 0.05).

### 3.3. Immunity Gene Expression

The expression of the *Pen3a* gene was significantly increased in the shrimp of the T4 group compared to the control and T5 groups (*p* < 0.05) (Figure 2). The expression of the *Toll* gene was significantly higher in the T3 group than in the T1 and T2 groups (*p* < 0.05). No differences were found in the expression of other immunity-related genes (*Lys*, *HSP70*, *Cru*, *ProPO*, and *IMD*) among all the groups (*p* > 0.05) (Figure 2).

### 3.4. Gut Microbiota

The removal of low-quality reads and chimeras yielded a total of 1,690,040 clean sequences with gut contents and lengths ranging from 401 to 440 bp (average 413 bp). Compared to the control and T4 groups, the Chao1 index was significantly increased in the shrimp of the T1 group (*p* < 0.05) (Figure 3). The ACE index was significantly lower in the T4 group than in the T1, T2, and T5 groups (*p* < 0.05). The Shannon index was significantly higher (*p* < 0.05) in the T5 group than in the control, T3, and T4 groups, and the lowest index was observed in the control group.

At the phylum level, the most dominant phyla in all the groups were Actinobacteria, Proteobacteria, and Firmicutes (Figure 4). Compared to the control group, the Proteobacteria abundance was significantly increased in the shrimp of the T4 and T5 groups (*p* < 0.05). At the genus level, the relative abundances of *Rhodobacter* and *Pseudahrensia* in the T2 group were significantly increased (*p* < 0.05) from those in the control group, but the *Exiguobacterium* abundance showed a reverse change trend (Figure 5). The relative abundance of *Hoeflea* in the T5 group was significantly higher than in the control group (*p* < 0.05). Furthermore, the beta diversity analysis between the different groups based on a PCA plot is shown in Figure 6. The gut microbiota structure of the shrimp in the T2 group was significantly increased from that of the control group.

The prediction functions were predicted using PICRUSt (https://docs.qiime2.org/2019.4/tutorials/, accessed on 5 December 2023). The ANOVA results showed that the KEGG pathways with significant differences between the T1 and T3 groups were divided into three categories at KEGG level 1: “metabolism”, “cellular processes”, and “organismal systems” (Table 4). In terms of nutrition-related metabolism functions, the carbohydrate metabolism, immune system, and environmental adaptation pathways were significantly higher in the T3 group than in the T1 group. In addition, the immune system and environmental adaptation pathways were also significantly higher in group T3.

## 4. Discussion

Findings from previous studies on the effects of β-glucan addition on the growth performance of shrimp were inconsistent in various aquaculture animals. Dietary β-glucan was found to enhance the growth performance of rainbow trout and shrimp [17,57,58]. However, other studies indicated that β-glucan did not promote the growth of Pengze crucian carp (*Carassius auratus var. Pengze*), rainbow trout, or Nile tilapia [59,60,61]. The results of the present study revealed that continuous feeding with β-glucan supplementation did not affect the growth performance for shrimp compared to a β-glucan-free diet. This suggests that continuous feeding with β-glucan did not act as a growth promoter in *L. vannamei* under low salinity. A further study also found that β-glucan could potentially play a substantial role in the efficiency of immunostimulation but did not increase the growth performance of rainbow trout [62]. Although continuous feeding with β-glucan did not promote the growth of shrimp, the intermittent feeding strategy resulted in a significantly higher condition factor, suggesting that intermittent dietary β-glucan feeding could promote growth under low salinity. Another study showed that alternating feeding of β-glucan could significantly enhance growth performance compared to continuous feeding in *L. vannamei* [31], which is consistent with the results of the present study. Therefore, these results suggest that the growth-promoting effect of β-glucan in aquatic animals is closely related to the feeding frequency under low salinity conditions.

β-glucan, especially yeast β-glucan, is considered one of the most efficient immune stimulants and has successfully enhanced the anti-infection ability of aquatic animals against viruses and bacteria [63,64]. It has been proven that β-glucan could improve the antioxidant activity of *L. vannamei* under low salinity stress [17]. T-AOC and GPX enzymes are two important antioxidant enzymes for scavenging oxygen free radicals [65,66]. The decline in T-AOC activity might be related to the higher ability of phagocytes to kill microorganisms [67]. According to this study, there was no significant difference in the T-AOC and GPX activities between the control and continuous feeding with the β-glucan group, indicating that continuous feeding of β-glucan did not improve the antioxidant activity of shrimp under low salinity. However, intermittent feeding with β-glucan resulted in decreased T-AOC and GPX activities, indicating that β-glucan promoted the antioxidant activity of shrimp, maintaining these two enzyme activities at low levels. It was noteworthy that the effectiveness of β-glucan was closely related to the use strategy [68]. The above results indicate that the effect of β-glucan on the antioxidant capacity of the host is closely related to the feeding strategy. Crustaceans rely on fixed nonspecific immune mechanisms, including coagulation cascades, antimicrobial peptides, reactive oxygen species, nitrogen intermediates, and antioxidant defense enzymes [69]. MDA is the end product of lipid peroxidation of polyunsaturated fatty acids and can be used to measure the extent of free radical damage to cells [70]. The results of the present study revealed that continuous β-glucan feeding had no impact on the MDA contents and did not significantly upregulate the mRNA expression of relevant innate immunity genes compared to the control group. These findings suggest that the continuous administration of β-glucan did not improve the immune ability of shrimp under low-salinity stress and even caused immune exhaustion in shrimp. To increase immunity in aquatic animals, it may be necessary to adjust the feeding strategy of β-glucan. The present study also revealed that the MDA content and expression level of the *Pen3a* and *Toll* genes in the hepatopancreas were significantly higher in the intermittent feeding strategy, indicating that an intermittent feeding strategy could enhance the innate immune function of shrimp. Similar observations have been reported, showing that feeding 0.2% of glucan in a specific feeding strategy resulted in a significantly higher immune response in shrimp [34]. In addition, a feeding strategy of once every week with yeast could increase immune response [71]. These outcomes may be attributed to the changed frequency of dietary alterations, suggesting that employing an intermittent feeding strategy of β-glucan can enhance both antioxidant and nonspecific immune responses, while avoiding immunosuppression.

The gut microbiota is very important in aquatic animal metabolism and nutritional balance, even regulating immunity and health [72,73,74,75]. Dietary β-glucan supplementation could influence the gut microbiota and gut barrier function of aquatic animals [17,60]. Furthermore, β-glucan administration could alter the structure of gut microbes by regulating the abundance of Lactobacilli, Bifidobacteria, Firmicutes, and Fusobacteria [76]. However, relatively small amounts of information are available on the alteration of gut microorganisms with different feeding strategies. In this study, it was found that the dominance of Proteobacteria was higher in the intermittent feeding strategy. A previous study showed that Proteobacteria is of supreme importance in the gut microbiota, which is sensitive to environmental factors such as diet, and that changes in relative abundance in the gut are related to the imbalance of gut microbiota [77]. At the genus level, a significant difference in the relative abundance of dominant bacteria was also observed. These findings suggest that a healthy gut microbiota in shrimp can be achieved through an optimum feeding strategy that includes dietary β-glucan. Further research into functional prediction could shed light on the importance of the gut microbiome in promoting host health [78,79,80]. This study suggests that intermittent feeding can effectively enhance carbohydrate metabolism in the gut microbiota. Specifically, intermittent feeding with β-glucan increases the production of short-chain fatty acids (SCFAs) and energy metabolism. In addition, KEGG pathways related to the immune system were enriched by intermittent feeding, indicating that this feeding strategy could increase the immune capability of shrimp. This finding can also explain the higher expression of immune-related genes in the intermittent feeding strategy. Thus, intermittent feeding might be a better approach to increase metabolism and prevent immune fatigue. Overall, the composition, diversity, and function of the gut microbiota in shrimp are significantly influenced by different feeding strategies.

## 5. Conclusions

In conclusion, different feeding strategies had a significant effect on *L. vannamei*. The intermittent feeding strategy of β-glucan could increase growth performance and antioxidant capacity. In addition, the intermittent feeding strategy of β-glucan could also increase the mRNA expression of the immunity gene and alleviate immune fatigue. Dietary β-glucan administered using an intermittent feeding strategy also affected the diversity and function of the gut microbiota. This study provides valuable information on the feeding strategies of β-glucan in *L. vannamei*.

## Figures and Tables

**Figure 1 animals-13-03778-f001:**
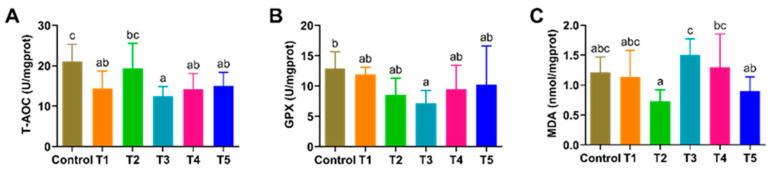
Effects of β-glucan on the antioxidant capacity of *L. vannamei* at different feeding frequencies. (**A**) Total antioxidative capacity (T-AOC), (**B**) glutathione peroxidase (GPX), and (**C**) malondialdehyde (MDA). Dissimilar letters show significant differences (*p* < 0.05). All the data are expressed as the mean ± SE (*n* = 4).

**Figure 2 animals-13-03778-f002:**
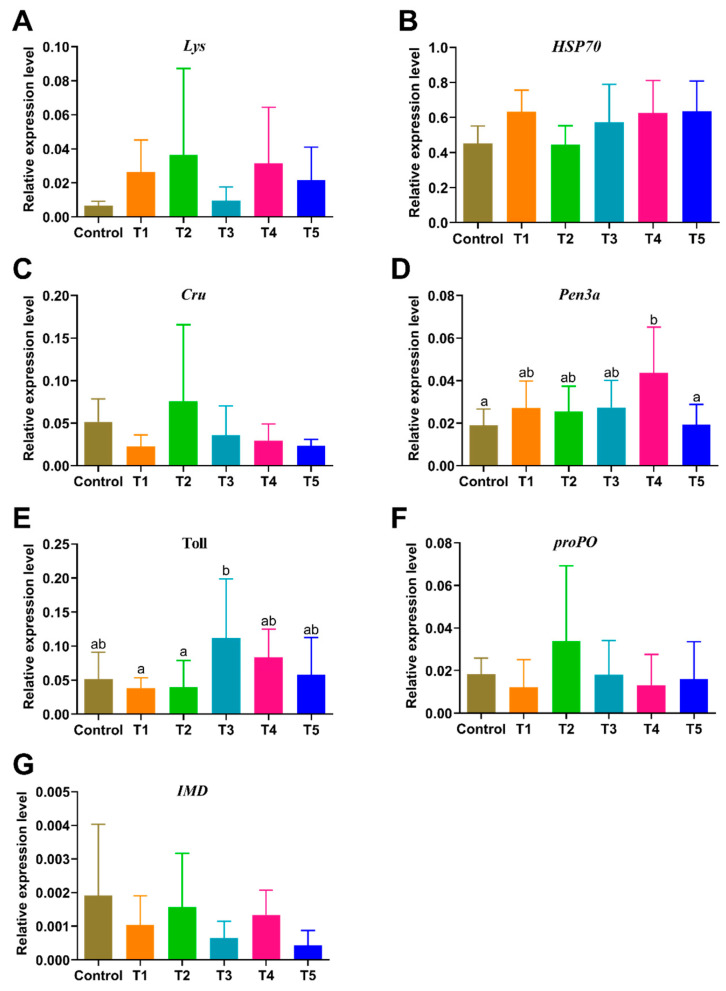
Effects of β-glucan on the expression levels of nonspecific immune-related genes in *L. vannamei* at different feeding frequencies. (**A**) Lysozyme (*Lys*), (**B**) Heat Shock Protein 70 (*HSP70*), (**C**) Crustin (*Cru*), (**D**) Penaeidins-3α (*Pen3a*), (**E**) Toll, (**F**) Prophenoloxidase (*ProPO*), and (**G**) immune deficiency (*IMD*). Dissimilar letters show significant differences (*p* < 0.05). All the data are expressed as the mean ± SE (*n* = 4).

**Figure 3 animals-13-03778-f003:**
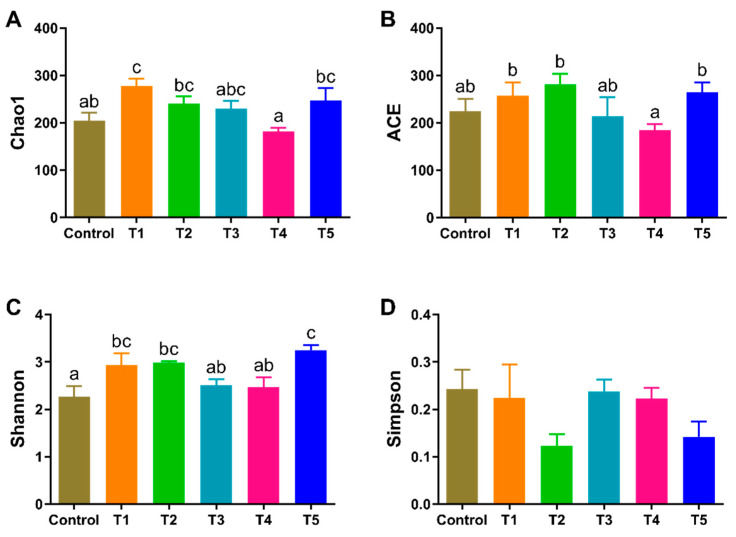
Effects of β-glucan on the alpha diversity of gut microbiota in *L. vannamei* at different feeding frequencies. (**A**) Chao1 estimator, (**B**) ACE estimator, (**C**) Shannon estimator, (**D**) Simpson estimator. Dissimilar letters show significant differences (*p* < 0.05). All the data are expressed as the mean ± SE (*n* = 4).

**Figure 4 animals-13-03778-f004:**
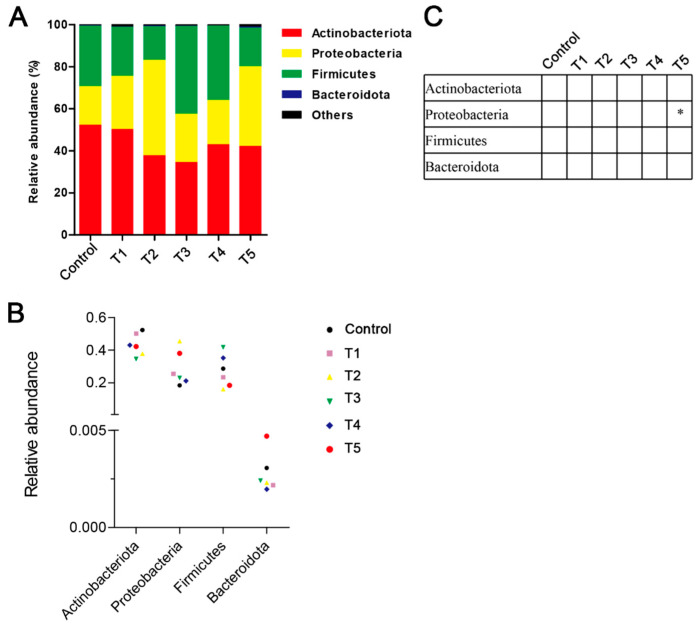
Effects of β-glucan on the gut microbiota composition at the phylum level in *L. vannamei* at different feeding frequencies. (**A**) Microbiota composition at the phylum level with relative abundance in the top four. (**B**) Relative abundance of gut microbiota at the phylum level. (**C**) Differences in the relative abundance of phylum taxa among the groups. The asterisks denote statistically significant differences * *p* < 0.05 compared to the control group.

**Figure 5 animals-13-03778-f005:**
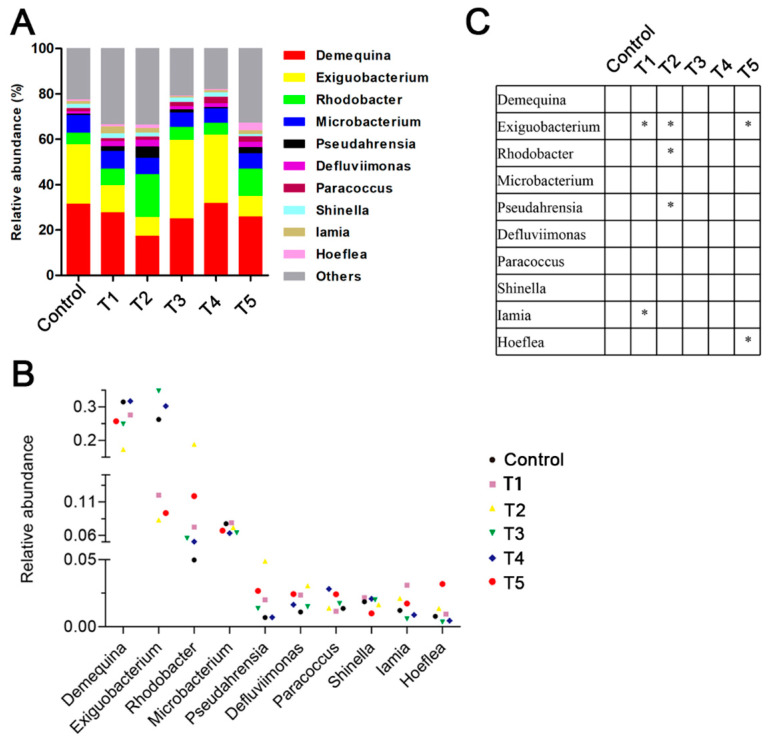
Effects of β-glucan on the gut microbiota composition at the genus level in *L. vannamei* at different feeding frequencies. (**A**) Microbiota composition at the genus level with relative abundance in the top ten. (**B**) Relative abundance of gut microbiota at the genus level. (**C**) Differences in the relative abundance of genus taxa among the groups. The asterisks denote statistically significant differences * *p* < 0.05 compared to the control group.

**Figure 6 animals-13-03778-f006:**
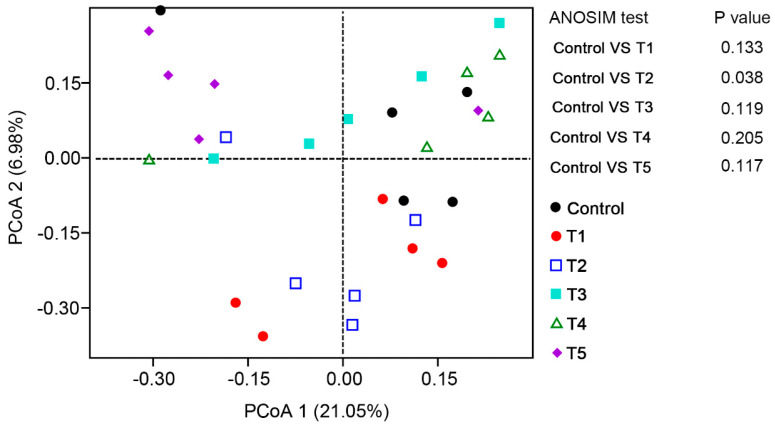
Effects of β-glucan on the β diversity of gut microbiota in *L. vannamei* at different feeding frequencies. PCoA of the microbiota at the OTU level based on Jaccard distances. Analysis of similarity (ANOSIM) tests were performed to evaluate the overall differences in bacterial community structure based on Jaccard distance.

**Table 1 animals-13-03778-t001:** Formulation and proximate composition of the experimental diets.

Ingredients (%)	Dietary β-(1, 3)-Glucan Concentration	
	Control	0.1% β-Glucan
Fish meal	26	26
Soybean meal	28	28
Corn starch	23	23
Shrimp meal	4	4
Calcium dihydrogen phosphate	1.5	1.5
Vitamin premix ^1^	2	2
Mineral premix ^2^	2	2
Choline chloride	1	1
Fish oil	2.5	2.5
Soybean oil	2.5	2.5
Soybean lecithin	1	1
Cholesterol	0.5	0.5
Carboxymethylcellulose	3	3
Butylated hydroxytoluene	0.1	0.1
Microcrystalline cellulose	2.9	2.8
β-(1, 3)-Glucan ^3^	0	0.1
Total	100	100
Nutrient levels (%)		
Crude protein	35.2	35.5
Crude lipid	7.6	7.6
Ash	10.3	10.7
Moisture	9.2	9.2

^1^ Vitamin premix (per kg of diet): vitamin A: 4800 IU; L-ascorbyl-2-polyphosphate 35% Active C: 35.71 g; folic acid: 0.18 g; biotin: 0.05 g; riboflavin: 3 g; DL Ca-pantothenate L: 5 g; pyridoxine HCl B_6_: 1 g; vitamin B_12_: 0.002 g; thiamine HCl: 0.5 g; Menadione K_3_: 2 g; DL-alpha-tocopheryl acetate: 20 IU; inositol: 5 g; nicotinamide: 5 g; and vitamin D: 8000 IU. ^2^ Mineral premix (per kg of diet): ZnSO_4_·H_2_O: 20.585 g; Ca (IO_3_)_2_: 0.117 g; CuSO_4_·5H_2_O: 0.625 g; MnSO_4_ H_2_O: 1.625 g; MgSO_4_·H_2_O: 39.86 g; CoCl_2_: 0.01 g; FeSO_4_·H_2_O: 11.179 g; and CaHPO_4_·2H_2_O: 166.442 g. ^3^ Yeast β-glucan was purchased from Xi’an Ruilin Biotechnology Co., Ltd., Xi’an, China.

**Table 2 animals-13-03778-t002:** Primers used for the real-time qPCR analysis ^1^.

Primer Name	Primer Sequence (5′ to 3′)	GenBank Accession	Tm	Product Size (bp)	Anneal	GC Content	References
*Lys*-F	GTTCCGATCTGATGTCCGATG	AY170126	56 °C	117	66 °C	52	[50]
*Lys*-R	AAGCCACCCAGGCAGAATAG					55	
*HSP70*-F	CCTCCAGGACTTCTTCAACG	AY645906	55 °C	145	63 °C	42	[51]
*HSP70*-R	GGTCACGTCCAACAGCAAC					53	
*Cru*-F	GAGGGTCAAGCCTACTGCTG	AY488497	57 °C	110	68 °C	60	[52]
*Cru*-R	ACTTATCGAGGCCAGCACAC					55	
*Pen3a*-F	CTCGTGGTCTGCCTGGTCTTCTTG	Y14926	58 °C	151	69 °C	58	[53]
*Pen3a*-R	CAGGGCAACCGTTGTATGGA					55	
*ProPO*-F	CGGTGACAAAGTTCCTCTTC	AY723296	54 °C	122	64 °C	50	[54]
*ProPO*-R	GCAGGTCGCCGTAGTAAG					61	
Toll-F	CCAGCTTAGAAGACCGGCAA	DQ923424	57 °C	83	68 °C	55	[55]
Toll-R	GTTGTCCGAGCAGAAGTCCA					55	
*IMD*-F	TGGGTCCGTGTCCAGTGATT	FJ592176	61 °C	79	70 °C	55	[55]
*IMD*-R	AGAGCCGCCGGTTATGTTGT					55	
*EF1α*-F	CCTATGTGCGTGGAGACCTTC	GU136229	56 °C	120	67 °C	57	[56]
*EF1α*-R	GCCAGATTGATCCTTCTTGTTGAC					46	

^1^ *ProPO*, prophenoloxidase; *Lys*, lysozyme; *Pen3a*, Penaeidins-3α; *Cru*, Crustin; *IMD*, immune deficiency; *HSP70*, Heat Shock Protein 70; *EF1α*, elongation factor 1 alpha.

**Table 3 animals-13-03778-t003:** The growth performance of *L. vannamei* at different β-glucan feeding frequencies.

β-Glucan Feeding Frequencies ^1^	Final Weight (g)	Weight Gain (%)	Specific Growth Rate (% day ^−1^)	Condition Factor (%)	Hepatosomatic Index (%)	Survival (%)
Control	7.65 ± 0.22	1502.55 ± 62.51	4.61 ± 0.07	1.02 ± 0.00 ^abc^	4.30 ± 0.07	95.00 ± 2.04
T1	8.02 ± 0.23	1521.07 ± 38.09	4.63 ± 0.04	1.01 ± 0.01 ^ab^	4.23 ± 0.08	96.25 ± 2.39
T2	7.56 ± 0.19	1477.29 ± 27.15	4.59 ± 0.03	1.04 ± 0.01 ^bc^	4.34 ± 0.09	95.00 ± 0.00
T3	8.31 ± 0.26	1543.33 ± 87.20	4.65 ± 0.09	1.05 ± 0.01 ^c^	4.15 ± 0.07	93.75 ± 2.39
T4	8.17 ± 0.10	1612.58 ± 91.25	4.71 ± 0.08	1.04 ± 0.01 ^bc^	4.09 ± 0.22	96.25 ± 2.39
T5	8.02 ± 0.30	1564.36 ± 100.23	4.66 ± 0.10	1.00 ± 0.01 ^a^	4.05 ± 0.12	96.25 ± 1.25

^1^ The values represent the means of four replicate tanks (*n* = 4). Different superscript small letters within a column indicate significant differences (*p* < 0.05).

**Table 4 animals-13-03778-t004:** Relative abundance of predicted microbial-mediated functions using PICRUSt. Abbreviations: T1 and T3 group (n = 4).

KEGG Level	KEGG Pathway	T1 (%)	T3 (%)	*p* Value
1	Metabolism			
2	Carbohydrate metabolism	12.382	12.570	0.042
3	Fructose and mannose metabolism	1.204	1.233	0.008
3	Galactose metabolism	0.365	0.426	0.018
2	Nucleotide metabolism	5.244	5.311	0.019
3	Purine metabolism	3.247	3.287	0.012
3	Cyanoamino acid metabolism	0.116	0.126	0.003
3	Polyketide sugar unit biosynthesis	0.098	0.099	0.001
3	Glycerolipid metabolism	0.335	0.377	0.006
3	Carbon fixation in photosynthetic organisms	0.464	0.473	0.017
1	Cellular Processes			
3	Focal adhesion	3.77 × 10^−6^	4.3 × 10^−6^	3.30 × 10^−2^
1	Organismal Systems			
2	Immune system	0.037	0.044	0.036
3	NOD-like receptor signaling pathway	4.23 × 10^−5^	8.55 × 10^−6^	0.08
3	RIG-I-like receptor signaling pathway	0.015	0.017	0.04
2	Environmental adaptation	0.200	0.204	0.018

## Data Availability

The data presented in this study are available on request from the corresponding author.

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
