# Peer review of "Effects of Dietary β-Glucan Feeding Strategy on the Growth, Physiological Response, and Gut Microbiota of Pacific White Shrimp, Litopenaeus vannamei, under Low Salinity"

_animals, 2023, doi:10.3390/ani13243778_

Round 1

Reviewer 1 Report

Comments and Suggestions for Authors

In the manuscript entitled " Effects of dietary β-glucan feeding strategy on the growth, physiological response, and gut microbiota of Pacific white shrimp, Litopenaeus vannamei, under low salinity", authors investigate the β-glucan feeding strategy on Pacific white shrimp (Litopenaeus vannamei) by supplementation after 6 weeks trial. The feeding trial was conducted to evaluate the effects of dietary β-glucan feeding strategy on the growth, physiological response, and gut microbiota, while there are too many writing specifications and grammatical errors in this manuscript. In addition, there also exists some inconsistencies between the detection indicators and the description of the results. This manuscript needs to be reorganized. 

My main issue is about the experiment:

1. In Simple summary, in Line 20, this statement is not correct “immunoenhancement”, meanwhile, please unify the expression between “gut” and “intestine”.

2. In line 36, the “α” in the Abstract should be used as the full name.

3. In line 41, the “change gut microbiota” is an incorrect statement, please revise it again.

4. In line 60, the reference format is wrong.

5. In line 110, the “12” should be subscript. Please verify and revise the full text of the manuscript.

6. In line 142, the shrimp cultured in the “tank”, not “group”.

7. In line 229, the shrimp treatment was named as “T3 group”, not “T3”. Please verify and revise the full text of the manuscript.

8. In line 280, this manuscript appears to be missing Table 4.

9. In line 370, the grammar should be present, not past.

Comments on the Quality of English Language

Minor editing of English language required

Author Response

1. In Simple summary, in Line 20, this statement is not correct “immunoenhancement”, meanwhile, please unify the expression between “gut” and “intestine”.

    Response: Thank you for highlighting these points, and the necessary revisions will be made to enhance the accuracy and coherence of the summary. (line 20)

2. In line 36, the “α” in the Abstract should be used as the full name.

    Response: Changed according to the comment. We replaced this symbol with its full name, “alpha.” (line 37)

3. In line 41, the “change gut microbiota” is an incorrect statement, please revise it again.

    Response: Changed according to the comment. We replaced this with “change gut microbiota composition”. (line 42)

4. In line 60, the reference format is wrong.

    Response: Changed according to the comment. We replaced this with “reference [13]”. (line 62)

5. In line 110, the “12” should be subscript. Please verify and revise the full text of the manuscript.

    Response: Changed according to the comment. (line 111-113)

6. In line 142, the shrimp cultured in the “tank”, not “group”.

    Response: Changed according to the comment. (line 142)

7. In line 229, the shrimp treatment was named as “T3 group”, not “T3”. Please verify and revise the full text of the manuscript.

    Response: Changed according to the comment. (line 246)

8. In line 280, this manuscript appears to be missing Table 4.

    Response: Added according to the comment. (line 319)

9. In line 370, the grammar should be present, not past.

    Response: Changed according to the comment. (line 394)

Reviewer 2 Report

Comments and Suggestions for Authors

My Decision on the manuscript with ID (animals-2713407-peer-review-v1) is “Major Revisions”. I have nine questions focused only on the Material and Methods section.

Q1. Line 95: Why did the authors select a dietary dose of 0.1% β-glucan?

Q2. Lines 120-122: Add detail about the acclimation of shrimp to low salinity. How long does it take?

Q3. Line 136: revise the feeding ratio used for shrimp throughout the experiment period

Q4. Lines 149-155: condition factor, hepatosomatic index, and survival rates are not related to “the growth performance -related parameters”.

Q5. Line 151: Add the equation for the calculation of weight gain (g) and add its results.

Q6. Line 168: Add a suitable reference for this method.

Benzie, I. F., & Strain, J. J. (1996). The ferric reducing ability of plasma (FRAP) as a measure of “antioxidant power”: the FRAP assay. Analytical Biochemistry, 239(1), 70-76.

Q7. Line 172 and 177: Add the code for the GPX and MDA kits used.

Q8. Line 177: Add a suitable reference for this method.

Asakawa, T., & Matsushita, S. (1979). Thiobarbituric acid test for detecting lipid peroxides. Lipids, 14(4), 401-406.

Q9. Line 200: Table 2 missed several important information such as

1. NCBI GenBank Accession numbers

2. Tm

3. Annealing temperatures

4. Primer efficiency

5. Product size (bp) 

Comments on the Quality of English Language

Moderate editing of English language required

Author Response

Q1. Line 95: Why did the authors select a dietary dose of 0.1% β-glucan?

    Response: The choice of the 0.1% β-glucan dose was informed by previous research conducted in 2020, which is cited for reference. The cited study demonstrated that the optimal level of dietary β-glucan dose under low salinity was 0.1%.

Qiao Y, Zhou L, Qu Y, Lu K, Han F, Li E. Effects of Different Dietary β-Glucan Levels on Antioxidant Capacity and Immunity, Gut Microbiota and Transcriptome Responses of White Shrimp (Litopenaeus vannamei) under Low Salinity. Antioxidants (Basel). 2022 Nov 18;11(11):2282.

Q2. Lines 120-122: Add detail about the acclimation of shrimp to low salinity. How long does it take?

    Response: The acclimation procedure for the shrimp to low salinity conditions in our study was carried out over a total of three weeks. Initially, we spent two weeks gradually reducing the salinity to a level of 3. Following this, one week was dedicated to ensuring that the shrimp could adequately adapt to these low salinity conditions.

Q3. Line 136: revise the feeding ratio used for shrimp throughout the experiment period。

    Response: Yes, during the feeding trail, we measured the shrimp weight from each tank per week and changed the feeding ratio.

Q4. Lines 149-155: condition factor, hepatosomatic index, and survival rates are not related to “the growth performance -related parameters”.

    Response: Changed according to the comment. (line 151)

Q5. Line 151: Add the equation for the calculation of weight gain (g) and add its results.

    Response: Changed according to the comment. (line 152)

Q6. Line 168: Add a suitable reference for this method.

Benzie, I. F., & Strain, J. J. (1996). The ferric reducing ability of plasma (FRAP) as a measure of “antioxidant power”: the FRAP assay. Analytical Biochemistry, 239(1), 70-76.‏

    Response: Changed according to the comment. (line 169)

Q7. Line 172 and 177: Add the code for the GPX and MDA kits used.

    Response: The GPX kit was code: A005-1-2 and MDA kit was code: A003-1-2.

Q8. Line 177: Add a suitable reference for this method.

Asakawa, T., & Matsushita, S. (1979). Thiobarbituric acid test for detecting lipid peroxides. Lipids, 14(4), 401-406.‏

    Response: Changed according to the comment. (line 178)

Q9. Line 200: Table 2 missed several important information such as

  1. NCBI GenBank Accession numbers
  2. Tm
  3. Annealing temperatures
  4. Primer efficiency
  5. Product size (bp) 

    Response: We have amended the table in accordance with the comments.  However, it is important to note that the primer sequences used in our experiment were referenced from other studies, and many of these sources did not provide detailed information about these aspects.  We appreciate the reviewer's valuable suggestion and plan to design our primers in future experiments, which will allow us to ensure the completeness of such information.

Reviewer 3 Report

Comments and Suggestions for Authors

The authors evaluated the effects of β-glucan supplemented diets on the growth performance, physiological response, and gut microbiota of Pacific white shrimp, assessing the expression of different antioxidant and immunity related genes and through a metagenomic analysis of the gut microbiota. Although the manuscript is well-written, my main concern goes to the experimental design and the M&M section, which I believe to be extremely inconsistent and lacking several important analysis. The authors should carefully read and follow the MIQE guidelines for the reporting of RT-qPCR data.

Comments:

Line 127 – why was no normal salinity-control group included?

Line 185 – no DNAse treatment was applied?

Line 187 – quality is hardly determined by NanoDrop, but rather by agarose gel or a bioanalyzer. Was this carried out?

Line 192 – Table 2 should contain annealing temperature, expected amplicon size and GC content for each primer. Please include this info here. Also, was the in silico specificity of constructed primers inspected?

Line 196 – were reactions performed in duplicate or triplicate? This should be indicated

Line 197 – was the specificity of the amplicons evaluated? Please indicate this here

Line 198 – according to the MIQE guidelines, a single reference gene is no longer accepted for RT-qPCR studies, and according to the M&M no expression stability analysis of this single gene was even carried out. I suggest to include at least one more reference gene

Line 210 – was this a paired-end or a single-end shotgun metagenomic study? Please detail

Line 212 – this line is repeated, the sequencing platform has been indicated in line 209.

Line 213 – which R package? This should be mentioned and correctly cited

Line 216 – why was the two-way ANOVA employed? Which two factors were considered? B-glucan and time? This should be specified

Line 213-219 – there is no description at all of all the bioinformatic analyses (e.g., raw reads preprocessing, taxonomic classification, functional prediction of the bacteriome, etc). This section is extremely incomplete 

Line 221 – which data do the authors refer here? Was this data transformed in any way prior to statistical analyses? Was normality and homocedasticity assessed?

Comments on the Quality of English Language

Minor check required

Author Response

Line 127 – why was no normal salinity-control group included?

    Response: This study was designed as a supplementary investigation to previously published research, focusing specifically on responses under varying low salinity conditions. Consequently, a normal salinity-control group was not included in the initial experimental design. We will certainly consider including a normal salinity-control group in our future experimental designs.

Line 185 – no DNAse treatment was applied?

    Response: Changed according to the comment. (line 187)

Line 187 – quality is hardly determined by NanoDrop, but rather by agarose gel or a bioanalyzer. Was this carried out?

    Response: Yes, we did.

Line 192 – Table 2 should contain annealing temperature, expected amplicon size and GC content for each primer. Please include this info here. Also, was the in silico specificity of constructed primers inspected?

    Response: Changed according to the comment. (line 205)

Line 196 – were reactions performed in duplicate or triplicate? This should be indicated

    Response: Changed according to the comment. (line 193)

Line 197 – was the specificity of the amplicons evaluated? Please indicate this here

    Response: The referenced literature did not provide data on the amplification efficiency of the primers.  However, our experimental results indicate that the primers functioned effectively in our specific assays.

Line 198 – according to the MIQE guidelines, a single reference gene is no longer accepted for RT-qPCR studies, and according to the M&M no expression stability analysis of this single gene was even carried out. I suggest to include at least one more reference gene

    Response: Thank you for pointing out this. We recognize that the inclusion of multiple reference genes, along with an expression stability analysis, is a best practice to ensure accuracy in quantitative gene expression studies. In our current study, we opted for a single reference gene due to specific constraints at the time of the experiment. However, we acknowledge this as a limitation of our study. Moving forward, we will incorporate your valuable suggestion of including two reference genes in our future experiments to align with the MIQE guidelines.

Line 210 – was this a paired-end or a single-end shotgun metagenomic study? Please detail

     Response: We have rewritten this section of the Materials and Methods to ensure that all the information is complete and clear.

Line 212 – this line is repeated, the sequencing platform has been indicated in line 209.

    Response: Changed according to the comment.

Line 213 – which R package? This should be mentioned and correctly cited

    Response: We have rewritten this section of the Materials and Methods to ensure that all the information is complete and clear.

Line 216 – why was the two-way ANOVA employed? Which two factors were considered? B-glucan and time? This should be specified

    Response: Changed according to the comment.

Line 213-219 – there is no description at all of all the bioinformatic analyses (e.g., raw reads preprocessing, taxonomic classification, functional prediction of the bacteriome, etc). This section is extremely incomplete 

    Response: We have rewritten this section of the Materials and Methods to ensure that all the information is complete and clear.

Line 221 – which data do the authors refer here? Was this data transformed in any way prior to statistical analyses? Was normality and homocedasticity assessed?

    Response: The statistical analysis used all statement data including Growth performance; Antioxidant capacity; Immunity gene expression; and the alpha diversity and composition of Gut microbiota. Prior to conducting statistical analyses, we rigorously assessed all raw data for normality and homoscedasticity. These assessments were carried out using standard statistical methods appropriate for each data type. Following this, we employed SPSS software for the statistical analysis. (line 221-223).

Round 2

Reviewer 2 Report

Comments and Suggestions for Authors

The authors correctly replied to the comments raised by the reviewer

Comments on the Quality of English Language

Minor editing of English language required

Reviewer 3 Report

Comments and Suggestions for Authors

Authors addressed the commments accordingly